# Post-Training Sparsity-Aware Quantization

**Gil Shomron**[†]  **Freddy Gabbay**[§]  **Samer Kurzum**[†]  **Uri Weiser**[†]

[†]Technion — Israel Institute of Technology, Haifa, Israel
[§]Ruppin Academic Center, Emek Hefer, Israel

{gilsho@campus, ssamer15@campus, uri.weiser@ee}.technion.ac.il
freddyg@ruppin.ac.il

## Abstract

Quantization is a technique used in deep neural networks (DNNs) to increase execution performance and hardware efficiency. Uniform post-training quantization (PTQ) methods are common, since they can be implemented efficiently in hardware and do not require extensive hardware resources or a training set. Mapping FP32 models to INT8 using uniform PTQ yields models with negligible accuracy degradation; however, reducing precision below 8 bits with PTQ is challenging, as accuracy degradation becomes noticeable, due to the increase in quantization noise. In this paper, we propose a sparsity-aware quantization (SPARQ) method, in which the unstructured and dynamic activation sparsity is leveraged in different representation granularities. 4-bit quantization, for example, is employed by dynamically examining the bits of 8-bit values and choosing a window of 4 bits, while first skipping zero-value bits. Moreover, instead of quantizing activation-by-activation to 4 bits, we focus on pairs of 8-bit activations and examine whether one of the two is equal to zero. If one is equal to zero, the second can opportunistically use the other's 4-bit budget; if both do not equal zero, then each is dynamically quantized to 4 bits, as described. SPARQ achieves minor accuracy degradation and a practical hardware implementation. The code is available at https://github.com/gilshm/sparq.

## 1   Introduction

Deep neural networks (DNNs) are at the heart of numerous applications, such as image classification and object detection [8], image synthesis [30], and recommendation systems [7]. DNNs, however, require abundant computations, as, for example, billions of multiply-and-accumulate (MAC) operations are required to assign a $224 \times 224$ colored image from the ImageNet dataset to one of its thousand possible classes. Limited computational resources, such as those in edge devices, latency constraints, and higher input resolutions, are all catalysts for development of methods that increase the ratio between DNN execution performance to hardware area, with as minimal impact on model accuracy as possible. One common method of doing so is quantization.

Quantization is commonly used to map the 32-bit floating-point (FP32) activations and weights in convolutional neural networks (CNNs) to 8-bit integers (INT8), which is known to result in minor or no degradation in model accuracy while easing hardware implementation [14]. Going below 8 bits, however, is not trivial, as quantization noise leads to a noticeable decrease in model accuracy. Quantization-aware training (QAT) methods employ training for quantization, to decrease quantization noise and recoup model accuracy [3, 25, 42]. Nevertheless, it is not always possible to employ training, for reasons such as lack of hardware resources, time, power, energy, dataset availability, or skilled manpower. Post-training quantization (PTQ) methods circumvent these issues [1, 5, 6].

35th Conference on Neural Information Processing Systems (NeurIPS 2021).

PTQ methods, basically, search for the optimal tensor clipping values to minimize quantization noise [1, 5]. They usually employ uniform quantization, since computing a dot product (DP) of evenly-spaced integer values can be implemented efficiently in hardware. DNN tensor distributions, however, are known to follow a bell-shaped distribution, such as Gaussian or Laplacian, i.e., the uniform quantization that is, on one hand, hardware-friendly, may not be, on the other hand, the best choice for minimizing the noise induced by the quantization process. To solve this mismatch, to some extent, PTQ methods that break tensor distributions into different quantization regions were proposed [6, 12, 24]. Computing a DP comprising values from different quantizations is not trivial though, since each activation-weight multiplication result may correspond to a different scaling factor, i.e., it will induce a multiplication by a different FP value per quantization region.

In this paper, we propose sparsity-aware quantization (SPARQ), which leverages the inherent and dynamic activation sparsity from granularities of entire integer 8-bit values (vSPARQ), down to INT8 representation zero-value bits (bSPARQ). With bSPARQ, instead of quantizing every activation to, for example, 4 bits according to a predetermined scaling factor, activations are first quantized to 8 bits and then dynamically quantized to 4 bits by choosing the most significant consecutive 4 bits while skipping leading zero bits (Figure 1). bSPARQ effectively achieves a number of quantization ranges while still enabling a practical hardware implementation.

Moreover, inspired by [32], we also leverage the entire 8-bit activation sparsity with vSPARQ, for additional mitigation of quantization noise. Instead of quantizing activation-by-activation to 4 bits, activations are quantized to 4 bits in pairs. If one activation is zero, then the other can span its bits across the first, and thereby still be represented by 8 bits to avoid additional quantization noise. If, however, both activations are non-zero, both are quantized to 4 bits by bSPARQ. We experiment with vSPARQ and bSPARQ in configurations of 4, 3, and 2 data bits.

This paper makes the following contributions:

- **Sparsity-aware quantization (SPARQ).** We present a sparsity-aware quantization method, in which $n$-bit quantization takes place by picking the most significant $n$ bits from the 8-bit value representation, while skipping leading zero-value bits. Moreover, since many activations are zero-value, we consider pairs of activations in the quantization process. If one activation is zero, the other can use the entire $2n$-bit budget. We experiment with a number of bit-group selection options and activation bit-widths that demonstrates the trade-off between model accuracy and hardware overhead.

- **Practical hardware implementation.** We implement SPARQ on top of a systolic array (SA), inspired by Google TPUs, and on top of a Tensor Core (TC) DP unit, inspired by NVIDIA GPUs, and show that SPARQ is practical in terms of area overheads. In addition, we also discuss SPARQ implementation on top of NVIDIA Sparse TCs (STCs), thus leveraging activation sparsity on top of weight sparsity.

- **Comprehensive evaluation.** We evaluate our method on a variety of image classification models, with numerous configurations and activation bit-widths, and compare it with previous PTQ works.

## 2   Related Work

PTQ methods are the most relevant works that are related to this work. ACIQ [1] analytically extracts the optimal quantization clipping values from the tensors' distributions and uses per-channel bit-allocation and per-channel quantization of activations. LBQ [5] formulates a minimum MSE optimization problem that is then solved numerically per layer, and employs additional low-precision tensors to sensitive layers. AdaQuant [10] and AdaRound [21] optimize the common round-to-nearest rounding scheme to reduce quantization noise. BRECQ [16] analyzes the second-order error and optimizes the quantization at block granularity. Conceptually, both vSPARQ and bSPARQ can be employed on top of any of the above quantizations (for simplicity's sake, we use a simple 8b-8b min-max symmetric quantization, as we also describe in Section 5).

Other works, such as OLAccel [24], PWLQ [6], and BiScaled-DNN [12], divide the tensor distribution into two regions. OLAccel divides the tensor distribution into a low-precision region that contains the majority of data, and a high-precision region that contains a small portion of the data (e.g., 3%), which they define as outliers. PWLQ and BiScaled-DNN, on the other hand, divide the tensor distribution

into two regions with the same bit-width. BiScaled-DNN uses different scale factors on overlapping regions and implements a ratio heuristic to set the breakpoint between the regions, whereas PWLQ picks the appropriate breakpoint via minimization of the quantization error. Interestingly, PWLQ is capable of breaking the distribution into more than two regions; however, the authors state that from a hardware perspective, this may not be feasible.

Following OLAccel, OverQ [41] leverages activation sparsity to avoid the dedicated outlier datapath used in OLAccel. In this work, we employ a simple rounding mechanism and bit-level sparsity to mitigate noise in the occasion a zero-value does not exist, and we propose a parallel implementation rather than a serial one.

SySMT [32] leverages sparsity in quantization of both activations and weights to 4 bits. Their method incurs relatively high area overheads, since the quantization logic has to be scaled with the number of processing units. Moreover, SySMT incurs relatively high degradation in accuracy, since quantization to 4 bits is implemented by trimming either the 4-bit most significant bits (MSBs) or the 4-bit least significant bits (LSBs). These two options are not optimal, since we find that, for example, with ResNet-18 and ILSVRC-2012, 67% of the non-zero-value activation values have at least one of the 4-bit MSBs toggled (i.e., equal to one), even though 90% of the time, the two MSBs are not toggled. That is, the two MSBs are most likely not toggled when the 4-bit MSBs are chosen.

# 3   The Basic Principle of SPARQ

SPARQ comprises two orthogonal techniques: bSPARQ and vSPARQ. The former leverages zero-value bits to trim an 8-bit value to an $n$-bit value; and the latter leverages zero-value activations. Below, we describe both in detail. Throughout this work, we focus on quantizing the activations and leveraging only their sparsity, i.e., no correlation is made with the weight values, unless otherwise stated.

## 3.1   bSPARQ: Leveraging Bit Sparsity

Consider an already quantized 8-bit activation, $x$, and quantization to 4 bits (i.e., $n = 4$). bSPARQ trims the activation from 8 bits to 4 bits by inspecting the activation bits and choosing the most significant consecutive 4 bits within it, which, in practice, is achieved by searching for the first most significant toggled bit. The motivation behind bSPARQ is twofold: first, activations usually follow a bell-shaped distribution, meaning that the MSBs are usually equal to zero and, therefore, can be skipped; and second, if the MSBs are toggled, the LSBs' contribution to the entire value is insignificant. For example, given the value $00011011_2$ ($27_{10}$), the 4-bit window will be positioned at bits [4:1] ($000\underline{1101}1_2$), thus achieving the approximated value $26_{10}$. Notice that since there are five window position options, the 4-bit window is accompanied by a 3-bit identifier that corresponds to the window position—that is, how much shift-left is required on top of the four trimmed bits. In addition, to further reduce the dynamic quantization noise, we round the value within the chosen window according to the residual LSBs. bSPARQ is visually demonstrated in Figure 1.

Supporting five window options requires additional circuitry compared with, for example, three window options, since additional placement options require additional hardware support by the shift-left unit. The trade-off is, however, improved accuracy, since additional placement options introduce less quantization noise. We experiment with five, three, and two placement options, denoted as 5opt, 3opt, and 2opt, respectively. With the 3opt configuration, [7:4], [5:2], or [3:0] are chosen, and with the 2opt configuration, either [7:4] or [3:0] are chosen (we leave the analysis of asymmetrical configurations for future work). For example, given the previous value, $00011011_2$, 3opt will choose bits [5:2] ($00\underline{0110}11_2$), whereas 2opt will choose bits [7:4] ($\underline{0001}1011_2$).

**Relation to piecewise linear quantization.** To mitigate quantization errors, previous works suggest dividing the tensor distributions into different quantization regions, each with a scaling factor of its own [6, 12, 24]. In a sense, bSPARQ is somewhat similar to those. First, each activation is assigned to a quantization range according to its value; however, we break the distributions into hardware-oriented regions of power of two. For example, for the 5opt case, the regions are $[0, 2^1 - 1]$, $[2^1, 2^2 - 1]$, and so on. As a result, values are mapped to their appropriate range by simply counting the leading zero bits. In addition, we avoid any need for preprocessing that searches for the distribution breakpoints to minimize the quantization noise. Second, each region has an individual scaling factor; however, each

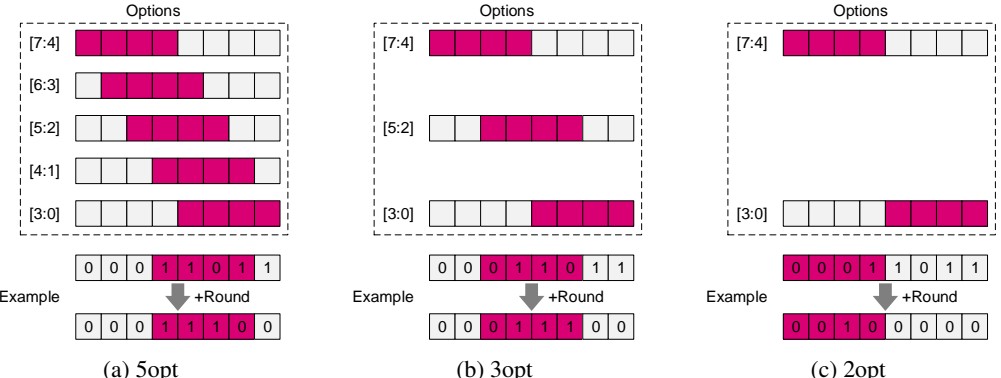

Figure 1: Demonstration of SPARQ 8b-to-4b quantization. More window placement options (e.g., 5opt) decrease the quantization noise; however, additional hardware is needed to support many placement options.

region scaling factor is a product of a base scaling factor with the corresponding power of two. For example, in the 5opt configuration, the scaling factor of the decimal number $33_{10} = 00\underline{100001}_2$ is the original scaling factor times $2^2$. This enables a relatively simple implementation with up to five regions when considering 4-bit activations, and even six and seven regions when considering 3- and 2-bit activations, respectively—as opposed to the two quantization regions used by previous works.

## 3.2 vSPARQ: Leveraging Sparsity with Pairs of Activations

Consider an 8-bit unsigned activation vector, $X = (x_1, \cdots, x_L)$, and an 8-bit signed weight vector, $W = (w_1, \cdots, w_L)$, both of length $L$. Also, consider a single MAC unit that computes a single activation-weight multiplication per cycle. vSPARQ, similar to [32, 34, 41], groups activations in pairs, to leverage the dynamic and unstructured activation sparsity. That is, the DP calculations can be formulated as:

$$X \cdot W = \sum_{i\,\text{even}}^{L} x_i w_i + x_{i+1} w_{i+1} = y, \tag{1}$$

where $y$ is the DP scalar result, and in our context, an output activation. For some $i$, if $x_i = 0$, then $x_{i+1}$ can be used with 8-bit representation, and vice versa. If, however, both $x_i \neq 0$ and $x_{i+1} \neq 0$, and given that, for example, bSPARQ is employed, then the precision of both $x_i$ and $x_{i+1}$ is reduced to 4 bits. For a certain $i$, the vSPARQ operation can also be formulated as:

$$x_i w_i + x_{i+1} w_{i+1} = \begin{cases} x_i w_i, & \text{if } x_{i+1} = 0 \\ x_{i+1} w_{i+1}, & \text{if } x_i = 0 \\ \text{bSPARQ}(x_i)w_i + \text{bSPARQ}(x_{i+1})w_{i+1}, & \text{otherwise} \end{cases}. \tag{2}$$

Notice that the two first case statements correspond to an 8b-8b computation, whereas the last case statement corresponds to two 4b-8b computations. The latter case is possible, since two 4b-8b multiplications are logically equivalent to a single 8b-8b multiplication, as we describe next.

**8b-8b = 2x4b-8b.** Given an 8-bit unsigned activation, $x$, and an 8-bit signed weight, $w$, the activation-weight multiplication can be formulated as

$$x_{[7:0]} \cdot w_{[7:0]} = \sum_{i=0}^{7} 2^i x_i \cdot w_{[7:0]} = \left( \sum_{i=0}^{3} 2^{i+4} x_{i+4} + \sum_{i=0}^{3} 2^i x_i \right) \cdot w_{[7:0]}$$
$$= 2^4 x_{[7:4]} \cdot w_{[7:0]} + x_{[3:0]} \cdot w_{[7:0]}, \tag{3}$$

where the $[b:a]$ notation represents the $b$-to-$a$ range in bits, the two activation-weight multiplications are 4b-8b wide, and the $2^4$ is equivalent to a 4-bit shift-left operation.

By considering an additional weight input as well as dynamic shift-left operations, we can reuse the multipliers and achieve a multiplier capable of either one 8b-8b multiplication or two *independent*

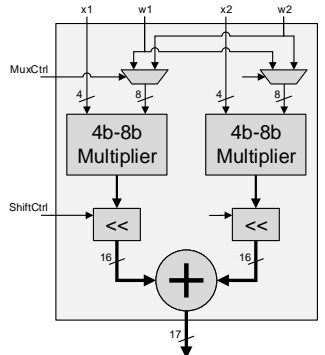 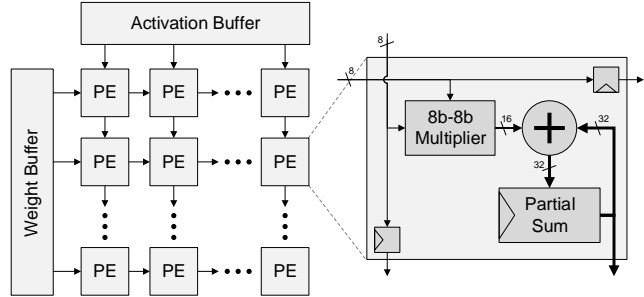

Figure 2: Equation (4) hardware implementation.

Figure 3: Illustration of a conventional 8b-8b output stationary systolic array.

4b-8b multiplications with a dynamic range:

$$2^{\text{opt}_1} x_{\text{in1,4b}} \cdot w_{\text{in1,8b}} + 2^{\text{opt}_2} x_{\text{in2,4b}} \cdot w_{\text{in2,8b}} , \tag{4}$$

where the activation and weight inputs are 4 bits and 8 bits long, respectively. Equation (4) resembles a FP representation; however, the "opt" configurations are not necessarily continuous, as in 3opt and 2opt. Figure 2 illustrates how Equation (4) is mapped to hardware. The two 4b-8b multipliers correspond to $x_{\text{in1}} \cdot w_{\text{in1}}$ and $x_{\text{in2}} \cdot w_{\text{in2}}$, and the two shift-left units ($\ll$) correspond to $2^{\text{opt}_1}$ and $2^{\text{opt}_2}$. The adder corresponds to the addition of the two groups, and the multiplexers, which are not explicitly formulated in Equation (4), are used to choose dynamically between $w_{\text{in1}}$, $w_{\text{in2}}$, or select both, during execution. We use this multiplier instead of the conventional one used in well-known hardware structures.

## 4 Case Studies

In this section, we examine SPARQ on top of two well-known matrix multiplication accelerator implementations: systolic arrays (SAs) and Tensor Cores (TCs). These accelerators are commonly used for CNNs, since it is a standard practice to map the convolution operation to matrix multiplication [2, 18, 39]. Our focus here is on the processing engines (PEs) comprising each of these structures and that are responsible for single DPs. Both implementations are fully equivalent from a mathematical point of view.

**Systolic arrays.** SAs consist of a large monolithic network of PEs designed for fast and efficient processing of systematic algorithms that execute the same computations with different data at different time instances [15]. The topology of SAs, illustrated in Figure 3, consists of a homogeneous network of tightly coupled PEs, each performing a MAC operation. PEs work in tandem: each PE in the SA receives data from its upstream neighbors, performs a MAC operation, and forwards the data downstream. In our PE design, also known as output-stationary SA, each PE will eventually hold the result of a DP; and the entire SA will comprise a tile from a result matrix. Google's TPUv2 and TPUv3, for example, consist of 128×128 SA arrays [22]. To deploy SPARQ, the conventional multiplier in each PE is replaced with the one presented in Figure 2, the weight bandwidth is doubled, and the activation bandwidth does not change.

**Tensor cores.** TCs were first introduced in NVIDIA's Volta architecture to accelerate matrix operations [4, 13, 19]. TCs multiply two 4×4 matrices and add an additional one to the multiplication result. The specific implementation details of TCs are not publicly disclosed; however, a proposed architecture that fits the original TC performance is suggested in [27]. In the proposed TC architecture, there are a number of DP units. Each DP unit performs four parallel activation-weight multiplications, accumulating them in an adder tree together with an additional third value. In this work, we focus on the architecture of a single DP, as presented in Figure 4. To enable SPARQ, the multipliers are replaced and the weight bandwidth is doubled, similar to the SA.

NVIDIA also recently introduced weight sparsity acceleration in its Ampere microarchitecture [20, 23]. The Sparse TC (STC) hardware achieves 2× speedup over the original TC by essentially

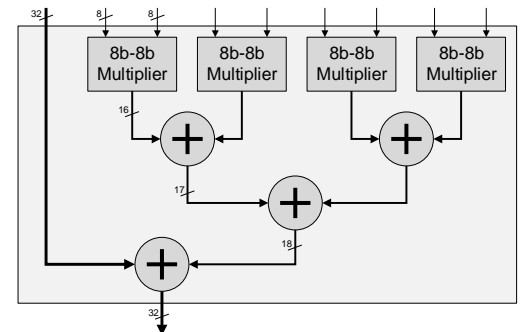

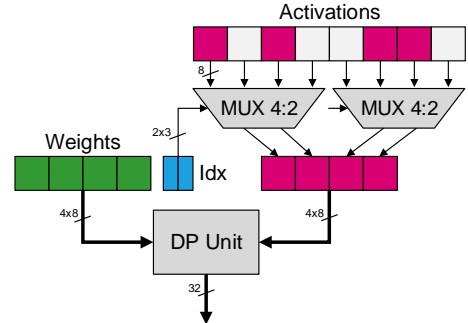

Figure 4: Illustration of a conventional 8b-8b DP unit comprising a TC [27].

Figure 5: Conventional STC microarchitecture [23].

skipping 50% of the computations (Figure 5). STC requires 50% weight structured pruning at a granularity of four elements, i.e., every four adjacent weights must have two zero-value weights. Only the non-zero-value weights are stored with additional coordinates. In Figure 5, the two leftmost weights and two rightmost weights correspond to the four leftmost activations and rightmost activations, respectively. The stored coordinates indicate which activations are picked, since they are to be multiplied by non-zero-value weights. After filtering the activations, they are passed with the weights to the DP unit for further processing. Notice, however, that activation sparsity may still exist even after the selection process.

## 5 Experiments

We evaluate the impact on model accuracy using PyTorch [26], the ILSVRC-2012 dataset [28], and various CNN models [8, 9, 11, 37, 37, 38] (see Table 1). All models are quantized using a simple uniform min-max quantization, employing symmetric unsigned per-layer quantization for activations and symmetric signed per-kernel quantization for weights. The min-max statistics are gathered during a quick preprocessing stage on 2K randomly picked images from the training set. In addition, during preprocessing, we recalibrate the BatchNorm layers' running mean and running variance statistics [29, 33, 35, 36]. In all models, the first convolution layer is left intact, since its input activations, which correspond to the image pixels, do not include many zero values, if any. Quantization is, therefore, performed on all convolution layers, with the exception of the first layer. We present the quantization results in Table 1 . Throughout this section, we use SPARQ on top of the 8-bit models (A8W8) and report the accuracy degradation relative to the corresponding FP32 model. A4W8 and A8W4 are presented in Table 1 as references to the worse-case accuracy.

Table 1: ILSVRC-2012 CNN top-1 accuracies, given different quantization precisions. Throughout this work, SPARQ is used on top of the A8W8 representation.

| Model | FP32 | A8W8 | A4W8 | A8W4 |
|---|---|---|---|---|
| ResNet-18 | 69.76% | 69.80% | 67.70% | 67.49% |
| ResNet-34 | 73.31% | 73.39% | 71.47% | 72.01% |
| ResNet-50 | 76.13% | 76.22% | 72.79% | 75.03% |
| ResNet-101 | 77.37% | 77.38% | 73.74% | 76.41% |
| GoogLeNet | 69.78% | 69.67% | 65.38% | 65.81% |
| Inception-v3 | 77.49% | 77.50% | 73.91% | 74.22% |
| DenseNet-121 | 74.69% | 74.68% | 72.57% | 72.89% |
| SqueezeNet | 58.09% | 57.81% | 28.12% | 34.14% |

In Section 5.3, we experiment with a 2:4 structured pruning [23]. To achieve the sparse model with the baseline accuracy, we prune the network based on its pretrained weights and retrain the model from scratch for 90 epochs with a learning rate starting from 0.1 and divided by 10 at epochs 30 and 60. Weight decay and momentum are set to 0.0001 and 0.9, respectively.

The different designs are implemented using SystemVerilog and synthesized using Synopsys® Design Compiler® and Virage (now Synopsys) 65nm standard cell library. We use a frequency of 500MHz at slow and fast corners for setup and hold timing closure, respectively. Area estimates were extracted after place-and-route using Cadence® Innovus™. We assume that the overall hardware overhead related to activation trimming and rounding is relatively negligible with respect to the SA and TC, since (1) the trimming and rounding unit involves a simple hardware scheme; and (2) it is performed at a significantly lower processing rate. We validated our multiplier against our PyTorch CUDA implementation with cycle-accurate testbenches to verify calculation integrity.

## 5.1 Accuracy Results

In Table 2, we present our method's results for the 5opt, 3opt, and 2opt configurations, with and without rounding ($\pm$R), as described in Section 3.1, and without vSPARQ (-vS). As expected, we observe that (1) better accuracy is achieved with the increase of window placement options; (2) overall, rounding further reduces quantization noise, which leads to smaller accuracy degradation; and (3) vSPARQ contribution is noticeable mainly in configurations with relatively high quantization noise. In addition, we observe a large impact on accuracy in the transition from 2opt to 3opt, since there is a high probability that at least one of the 4-bit MSBs will be toggled. For example, given the non-zero-valued activations in ResNet-18 with the ILSVRC-2012 dataset, we measure that bits 7, 6, 5, and 4 are toggled in 0.5%, 9.2%, 33.8%, and 44.8% of the time, respectively. Assuming the bit values are statistically independent, the probability of at least one toggled bit is 67%. Notice that there is a clear redundancy in the 2opt configuration that picks the 4-bit MSBs, even though 10% of the time the two MSBs are toggled.

Table 2: SPARQ accuracy results using the ILSVRC-2012 dataset, without rounding (-R), with rounding (+R), and with rounding but without vSPARQ (+R-vS).

| | 5opt | | | 3opt | | | 2opt | | |
|---|---|---|---|---|---|---|---|---|---|
| Model | Trim | +R | +R-bS | Trim | +R | +R-bS | Trim | +R | +R-bS |
| ResNet-18 | -0.11% | -0.07% | -0.11% | -0.22% | -0.14% | -0.48% | -2.87% | -1.37% | -2.02% |
| ResNet-34 | -0.00% | +0.04% | -0.05% | -0.25% | -0.14% | -0.25% | -2.38% | -1.10% | -1.75% |
| ResNet-50 | -0.03% | -0.05% | -0.02% | -0.41% | -0.18% | -0.31% | -4.18% | -2.18% | -2.83% |
| ResNet-110 | -0.22% | -0.25% | -0.19% | -0.67% | -0.59% | -0.60% | -3.31% | -1.64% | -2.82% |
| GoogLeNet | -0.83% | -0.68% | -0.77% | -1.59% | -0.75% | -0.99% | -5.14% | -2.55% | -4.31% |
| Inception-v3 | -0.73% | -0.62% | -0.95% | -1.51% | -1.21% | -1.68% | -3.98% | -1.86% | -3.30% |
| DenseNet-121 | +0.10% | +0.09% | +0.05% | -0.16% | +0.05% | -0.02% | -2.39% | -0.57% | -1.10% |
| SqueezeNet | -1.63% | -0.80% | -0.90% | -3.73% | -1.05% | -1.26% | -54.5% | -8.24% | -11.6% |

Computationally, SPARQ may be considered as a dynamic 4b-8b PTQ, in which quantization to 4 bits from 8 bits is conducted occasionally in the event of two adjacent non-zero-value activations. The upside of conventional PTQ methods, however, is the reduction in memory footprint, where the dynamic method falls short, due to the additional metadata. For example, the 3opt configuration requires additional 3-bit metadata per 4-bit activation data (2-bit ShiftCtrl and 1-bit MuxCtrl). Still, the memory footprint may be reduced by grouping the metadata for several activations, which we leave for future exploration. In Table 3, we present our results compared with previous related works [1, 5, 6, 31]. We would like to point out that SySMT is similar to the 2opt configuration. The slightly different results are due to the different BatchNorm calibrations and the slightly different 8-bit quantized models. Regarding ResNet-50, SySMT quantizes its weights, whereas SPARQ focuses on quantizing activations.

**Reducing the bit width: 3 bits and 2 bits.** To further challenge SPARQ efficiency, we experiment with 3-bit and 2-bit configurations. The lower budget leads to increased quantization noise even when one of the activations within the activation pair has a zero value, since the total window sizes are 6 and 4 bits for the 3-bit and 2-bit configurations, respectively. In Table 4, we present SPARQ accuracy results compared with other methods that reported sub-4b quantization results. As opposed to Table 2, we observe that vSPARQ impact is more significant in lower bit-widths.

Table 3: Relative top-1 accuracy degradation (relative to FP32) of SPARQ versus different quantization methods used for 4b-8b quantization (the best out of 4-bit activations or weights).

| Model | SPARQ | | | SySMT | PWLQ | ACIQ | LBQ | KURE |
| | 5opt | 3opt | 2opt | | | | | |
|---|---|---|---|---|---|---|---|---|
| ResNet-18 | -0.07% | -0.14% | -1.37% | -1.29% | - | -2.01% | -1.20% | -2.84% |
| ResNet-34 | +0.04% | -0.14% | -1.10% | - | - | - | - | - |
| ResNet-50 | -0.03% | -0.18% | -2.18% | -0.43% | -0.67% | -1.05% | -1.36% | -0.92% |
| ResNet-101 | -0.22% | -0.59% | -1.64% | - | - | -0.52% | -1.18% | - |
| GoogLeNet | -0.68% | -0.75% | -2.55% | -2.85% | - | - | - | - |
| Inception-v3 | -0.62% | -1.21% | -1.86% | - | -1.34% | -2.72% | -1.88% | - |
| DenseNet-121 | +0.10% | +0.05% | -0.57% | -0.39% | - | - | -1.17% | - |
| SqueezeNet | -0.80% | -1.05% | -8.24% | - | - | - | -2.96% | - |

Table 4: Relative top-1 accuracy degradation (relative to FP32) for 3-bit and 2-bit SPARQ (with 8-bit weights) in 6opt and 7opt configurations, respectively, also with and without vSPARQ (-vS).

| Model | SPARQ | | | | KURE | | ACIQ |
| | 3b | 2b | 3b (-vS) | 2b (-vS) | 3b | 2b | 3b |
|---|---|---|---|---|---|---|---|
| ResNet-18 | -0.21% | -1.64% | -0.51% | -2.57% | -10.9% | -42.8% | -17.1% |
| ResNet-34 | -0.18% | -1.19% | -0.37% | -1.66% | - | - | - |
| ResNet-50 | -0.59% | -2.34% | -0.73% | -3.53% | -3.53% | -15.9% | -11.4% |
| ResNet-101 | -0.66% | -2.64% | -1.06% | -3.73% | - | - | -6.08% |
| GoogLeNet | -1.32% | -6.47% | -1.91% | -9.16% | - | - | - |
| Inception-v3 | -1.70% | -5.60% | -2.45% | -9.29% | - | - | -26.4% |
| DenseNet-121 | -0.07% | -0.86% | -0.25% | -1.73% | - | - | - |
| SqueezeNet | -1.63% | -10.4% | -2.32% | -15.0% | - | - | - |

## 5.2 Hardware Evaluation

Table 5 summarizes the area overhead normalized to the MAC throughput of SPARQ for both SA and TC use cases. The SA and TC baselines are conventional 8b-8b SA and TC PEs, respectively. Memory, such as SRAMs, are not considered in the analysis (which could decrease the area overhead percentages). The 2×4b-8b design is presented as a reference implementation in the case of 4b-8b quantized values with equivalent throughput to the design in Figure 2. For the sake of fair comparison, there is a single psum in the 2×4b-8b design.

With respect to the SA, the 2×4b-8b PE requires approximately half the area per MAC operation than the 8b-8b PE. On the one hand, the total multipliers' area of the 2×4b-8b PE is significantly smaller; however, the 2×4b-8b PE employs a 3-input adder. The shift-left logic is the main contributor to the increasing area overhead of opt2 through opt5. As the number of shift-left options increases, the shift logic becomes more complex and utilizes a bigger logic area. Regarding 6opt (3 bits) and 7opt (2 bits) configurations, even though they require additional window placement options, the overall area decreases, since the area of the multipliers, registers, and multiplexers within the shift-left units is reduced. Also, our 2opt scheme introduces a significantly smaller area overhead compared with SySMT, due to the fact that SySMT required the trimming and rounding hardware to operate at the same high throughput rate as the SA. Regarding TC, the 2×4b-8b implementation requires half the area (normalized) of the TC 8b-8b baseline PE. Similar to the SA use case, the 2×4b-8b PE multipliers are smaller; however, this time the 2×4b-8b PE adder tree grows.

Interestingly, the relative area of 5opt no-vSPARQ (-vS) is only slightly higher than the "full" 3opt SPARQ implementation. Given the accuracy differences between the two configurations (Table 2), the 3opt SPARQ operating point presented in this work may not be a good trade-off between accuracy and area.

Table 5: Relative hardware area (normalized to MAC operation throughput) of different SA and TC implementations.

| Type | 8b-8b | 2×4b-8b | SPARQ | | | | | SPARQ (-vS) | | SySMT |
|---|---|---|---|---|---|---|---|---|---|---|
| | | | 7opt | 6opt | 5opt | 3opt | 2opt | 5opt | 3opt | |
| Systolic Array PE | 1.00 | 0.50 | 0.59 | 0.66 | 0.72 | 0.61 | 0.57 | 0.62 | 0.59 | 0.72 |
| Tensor Core PE | 1.00 | 0.50 | 0.58 | 0.63 | 0.72 | 0.66 | 0.61 | 0.67 | 0.61 | - |

Table 6: Accuracy results of SPARQ simulated on top of an STC with 2:4 structured pruned models.

| Model | FP32 | A8W8 | 4-bit | | | 3-bit | 2-bit |
|---|---|---|---|---|---|---|---|
| | | | 5opt | 3opt | 2opt | 6opt | 7opt |
| ResNet-18 | 69.77% | 69.79% | -0.13% | -0.34% | -1.59% | -0.41% | -1.92% |
| ResNet-50 | 76.16% | 76.10% | -0.24% | -0.57% | -2.59% | -0.85% | -3.18% |
| ResNet-101 | 77.38% | 77.34% | -0.28% | -0.39% | -2.06% | -0.79% | -2.94% |

### 5.3 Leveraging Activation Sparsity on Top of Sparse Tensor Cores

We simulate SPARQ on top of an STC with models pruned with 2:4 structured pruning. As presented in Figure 5, activations are first filtered through the multiplexers according to the non-zero-value weight coordinates. Then, vSPARQ comes into play, inspecting pairs of activations, as described in Section 3. Since in STC the trimming and rounding logic should be replicated for each DP unit, we implemented and synthesized the trimming and rounding unit to estimate its area overhead. The unit area, relative to the conventional TC (Figure 4), is 17%, 12%, and 9% for the 5opt, 3opt, and 2opt configurations, respectively. The relative area may be even smaller if we consider the entire STC design (Figure 5). SPARQ is, therefore, beneficial in terms of performance-to-area when attached to an STC.

In Table 6, we report the pruned models' FP32 and A8W8 quantized accuracies, and repeat all experiments described thus far. Interestingly, the relative accuracy degradation of the pruned models is slightly higher than that of the unpruned models in Table 3 [17, 40]. Nevertheless, SPARQ still achieves less than 1% relative degradation in accuracy with 4-bit 5opt and 3opt, and 3-bit 6opt.

## 6 Limitations and Societal Impacts

SPARQ has two main limitations: (1) It does not achieve the memory footprint decrease as native 4b-8b quantization methods do, because of the additional metadata that accompanies each value, as discussed in Section 5.1. The memory footprint may be decreased by giving up vSPARQ or sharing ShiftCtrl for a number of activations. We leave these research directions for future work. (2) From a hardware perspective, SPARQ requires hardware support, i.e., it cannot run on today's commodity hardware. In addition, compared with native 4b-8b quantizations, our hardware implementation incurs some overhead, as described in Section 5.2.

As for the societal impacts, quantization methods, in general, increase the effective amount of available computing resources, since the execution requirements of quantized models are lower. The effective increase in computing power may be targeted towards negative use, such as surveillance and fake profile generation.

## 7 Conclusion

We present SPARQ, a sparsity-aware quantization method that dynamically leverages sparsity in different granularities—from the entire 8-bit value to the individual bits. Thanks to the inherent activation sparsity, quantization to $n$ bits occurs only occasionally. When quantization to $n$ bits does occur, bit-level sparsity is leveraged by trimming leading zero bits and picking the most significant consecutive $n$ bits. SPARQ induces minor accuracy degradation and is hardware-friendly.

## Acknowledgements

We thank the anonymous reviewers for their comments and suggestions. We also thank Moran Shkolnik, Mario Shalabi, and Michael Behar for their valuable feedback.

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
