# OpenReview forum: "Post-Training Sparsity-Aware Quantization"
_NeurIPS.cc/2021/Conference — NeurIPS 2021 Poster_

### Official Review · Reviewer_DrG4 · 2021-07-15

**Rating:** 6
**Confidence:** 3

**Summary:**

This paper proposes a sparsity-aware quantization (SPARQ) method, in which n-bit quantization takes place by picking the most significant n bits from the 8-bit value representation. Also, SPARQ is implemented on a systolic array(SA) or a tensor core(TC) DP unit. Finally, compared with the previous PTQ work, this paper evaluates the quantization effect of this method for various image classification models, and achieves the most advanced results.

**Ethics Review Area:**

["I don’t know"]

**Limitations And Societal Impact:**

- According to the description of the paper, is the hardware implementation only at the algorithm level?
- Please describe the process of 8b-4b in Figure 1 with words and figures more clearly.
- The results shown in Table 1,2,4 are not clear enough. Can the full precision results be reflected in tables or words?
- Compared with other methods, the accuracy of the results presented in this paper is greatly improved, and the description of hardware implementation is surprising.


**Main Review:**

- According to the description of the paper, is the hardware implementation only at the algorithm level?
- Please describe the process of 8b-4b in Figure 1 with words and figures more clearly.
- The results shown in Table 1,2,4 are not clear enough. Can the full precision results be reflected in tables or words?
- Compared with other methods, the accuracy of the results presented in this paper is greatly improved, and the description of hardware implementation is surprising.


**Time Spent Reviewing:**

11

---

> ### Author Response · Authors · 2021-08-07
> **Response to Reviewer DrG4**
>
> We would like to thank the reviewer for the helpful and valuable feedback and suggestions. We would like to address the reviewer comments:
>
>
> * Q: “According to the description of the paper, is the hardware implementation only at the algorithm level?”
> A: The different designs are implemented using SystemVerilog, synthesized using Synopsys Design Compiler, and placed-and-routed using Cadence Innovus. We used Virage 65nm standard cell library. The experimental setup is described in the supplementary material, but we will try and squeeze it into the main paper.
>
>
> * Q: “Please describe the process of 8b-4b in Figure 1 with words and figures more clearly.”
> A: We thank the reviewer for this feedback. We will go over our explanations in Section 3.1, and will describe Figure 1 more clearly.
>
>
> * Q: “The results shown in Table 1,2,4 are not clear enough. Can the full precision results be reflected in tables or words?”
> A: The absolute model accuracies are presented in the supplementary material, and the related work absolute model accuracies can be found in the original papers. We can try to redesign the tables to include the absolute values as well.

---

### Official Review · Reviewer_8ZWQ · 2021-07-16

**Rating:** 5
**Confidence:** 4

**Summary:**

This paper presents a sparsity-aware post-training quantization approach, which considers the sparsity on both the digit representation level and the activation level. The algorithm thus can select the most significant bits from the original 8-bit value representation, and dynamically adjust the bit budget allocation for the activation pairs during multiplication. The authors also discuss the hardware implementation for the proposed approach based on systolic array and Tensor Core DP unit. Experimental results show the effectiveness of the proposed method on various network architectures. Nonetheless, the paper still has some unclear issues with the experimental results and hardware implementations.

**Limitations And Societal Impact:**

The authors have adequately addressed the limitations and potential negative societal impact of their work.

**Main Review:**


Strengths:

* The paper proposes a novel solution that leverages the sparsity of bit representations and activations to enhance the quantized representation dynamically.

* Comprehensive empirical evaluations are provided to verify the proposed approach.

* The paper is overall easy to follow.

Weakness:

* The paper has missing some recent efforts on post-training quantization listed below [1,2,3,4]. These works largely follow uniform quantization and are therefore friendly to general hardwares.

  [1] Hubara I, Nahshan Y, Hanani Y, et al. Improving post training neural quantization: Layer-wise calibration and integer programming. ICML, 2021.

  [2] Li Y, Gong R, Tan X, et al. Brecq: Pushing the limit of post-training quantization by block reconstruction. ICLR 2021.

  [3] Nagel M, Amjad R A, Van Baalen M, et al. Up or down? adaptive rounding for post-training quantization. ICML, 2020.

  [4] Nahshan Y, Chmiel B, Baskin C, et al. Loss aware post-training quantization. arXiv preprint arXiv:1911.07190, 2019.

* The hardware implementation and evaluation are not clear enough:

  - In terms of the hardware evaluation, please introduce more details on how to calculate the hardware area for the proposed method, which would be clear to the general audience.

  - It is still unclear why the proposed method enjoys 2x speedup. Is it due to the design of STC (L195)? Does the performance refer to acceleration in L251? What about the time consumption on hardware to search the most significant bits in bSPARQ (L111)? Do you measure the practical hardware speed up for your proposed algorithm?

* More issues for the experiments:

  - What is the absolute value accuracy of the quantized models? For now the results are mostly based on the accuracy degradation, however, the original model might have different full-precision accuracies in the first place.

  - It would be more comprehensive to compare with the missing recent works [1,2,3,4]? Especially, BRECQ[2] also considers the low bit post-training quantization on ResNet-18 and ResNet-50. It would be also convenient to compare with these approaches with absolute accuracy values reported.

  - Missing ablations studies for results without bSPRAC or vSPARC. You may present the ablation for low bit quantization (e.g. Table 5), so that the gain of bSPARC and vSPARC can be presented more clearly.



Detailed comments:

* It would be more inspiring to show the statistics of activation sparsity for different layers or network architectures. This illustrates how much we can benefit from vSPARQ (Equation 2).

* Is the proposed method only designed for activation quantization? What are the potential issues if applied for weight quantization?

* The bit sparsity under sparsity-aware post-training quantization seems kind of misleading to me, which simply refers to the zeros in the binary digits of 8-bit representation.

* For bSPARQ, what about trimming the bit-width with non-consecutive but equally spanned digits, e.g., 4-bit at 1,3,5,7 or 2,4,6,8 digits from the 8-bit representation?


**Time Spent Reviewing:**

4

---

> ### Author Response · Authors · 2021-08-07
> **Response to Reviewer 8ZWQ**
>
> We would like to thank the reviewer for the helpful and valuable feedback and suggestions. We would like to address the reviewer comments about the paper’s weaknesses:
>
>
> * Q: "The paper has missing some recent efforts on post-training quantization listed below [1,2,3,4]. These works largely follow uniform quantization and are therefore friendly to general hardwares."
> A: Thank you for these pointers. Specifically, LAPQ [4] does not employ per-channel quantization, which makes the comparison not fair.
> As for the rest of the papers, we will check them and add them to our comparison table.
>
>
> * Q: "In terms of the hardware evaluation, please introduce more details on how to calculate the hardware area for the proposed method, which would be clear to the general audience."
> A: The different designs are implemented using SystemVerilog, synthesized using Synopsys Design Compiler, and placed-and-routed using Cadence Innovus. We used Virage 65nm standard cell library. The experimental setup is described in the supplementary material, but we will try and squeeze it into the main paper.
>
>
> * Q: "It is still unclear why the proposed method enjoys 2x speedup. Is it due to the design of STC (L195)? Does the performance refer to acceleration in L251? What about the time consumption on hardware to search the most significant bits in bSPARQ (L111)? Do you measure the practical hardware speed up for your proposed algorithm?"
> A: The potential speedup of, for example, 4b-8b models over 8b-8b models is 2x, given the same area, give or take, since an 8b-8b multiplication is equivalent to two 4b-8b multiplications (Equation 3). One can potentially achieve 2x speedup over the 8b-8b implementation with a 4b-8b model running on a 2x4b-8b, SPARQ, or SySMT architecture, with the area overheads presented in Table 3.
>
>
> * Q: "What is the absolute value accuracy of the quantized models? For now the results are mostly based on the accuracy degradation, however, the original model might have different full-precision accuracies in the first place."
> A: The absolute model accuracies are presented in the supplementary material, and the related work absolute model accuracies can be found in the original papers. Throughout this work, we calculate the relative accuracy degradations with their corresponding baseline accuracies, since, as the reviewer mentioned, different works may use slightly different baselines.
>
>
> * Q: "It would be more comprehensive to compare with the missing recent works [1,2,3,4]? Especially, BRECQ[2] also considers the low bit post-training quantization on ResNet-18 and ResNet-50. It would be also convenient to compare with these approaches with absolute accuracy values reported."
> A: Thank you for that pointer. We will gladly add BRECQ to our comparison tables. For example, BRECQ achieves relative accuracy degradation of 0.53%, 1.79%, and 6.72% for ResNet-18 with 4/32, 3/32, and 2/32, respectively, whereas SPARQ achieves 0.07%, 0.21%, and 1.64%.
>
>
> * Q: "Missing ablations studies for results without bSPRAC or vSPARC. You may present the ablation for low bit quantization (e.g. Table 5), so that the gain of bSPARC and vSPARC can be presented more clearly.”
> A: We thank the reviewer for this suggestion. We enabled an additional argument in our code which disables vSPARQ, and measured all operating points with only bSPARQ. We will gladly add the numerical results and conclusions into a revision and update our code in the repository. These are our conclusions:
>     * With 5opt, vSPARQ impact is minor, since bSPARQ incurs relatively small quantization noise. If we remove vSPARQ from our hardware implementation (the two top multiplexers and MuxCtrl in Figure 2), we measure 10% area overhead decrease for the 5opt implementations in Table 4.
>     * Given 5opt results, the 3opt configuration becomes obsolete, since for almost the same hardware, 5opt achieves better accuracies.
>     * vSPARQ comes into play for 2opt and lower bit widths (Table 5), in which the relatively high quantization noise of a quantized element can be mitigated by leveraging sparsity. For example, ResNet-101 top-1 accuracy with 2-bit vSPARQ is 75.33%, whereas without vSPARQ it is 74.49%.
>
>
> * Q: "It would be more inspiring to show the statistics of activation sparsity for different layers or network architectures. This illustrates how much we can benefit from vSPARQ (Equation 2)."
> A: We agree with the reviewer, it is indeed an interesting statistic. It is known that sparsity exists in the different layers [5, 6]. However, as seen in the ablation study, it is not only how many 8b-to-4b quantizations are avoided, but also which are avoided. For example, the 5opt method does not incur much quantization noise to begin with; therefore, vSPARQ does not contribute much in this configuration. We will definitely discuss this if we'll be given a chance for a revision.
> [5] Parashar, Angshuman, et al. "SCNN: An Accelerator for Compressed-Sparse Convolutional Neural Networks." ISCA, 2017.
> [6] Gong, Zhangxiaowen, et al. "SparseTrain: Leveraging Dynamic Sparsity in Software for Training DNNs on General-Purpose SIMD Processors." PACT, 2020.
>
>
> * Q: "Is the proposed method only designed for activation quantization? What are the potential issues if applied for weight quantization?
> A: Both bSPARQ and vSPARQ can be applied to weights, as well as on activations. vSPARQ, however, will only contribute something if weights accommodate zeros, which is why we focused on activations that inherently comprise unstructured sparsity. Exploring the impact of bSPARQ on weights is definitely a future work - we thank the reviewer for this idea.
>
>
> * Q: "The bit sparsity under sparsity-aware post-training quantization seems kind of misleading to me, which simply refers to the zeros in the binary digits of 8-bit representation."
> A: We argue that the novelty of this work is that sparsity can be leveraged for quantization in the entire representation granularity (vSPARQ) as well as on bit-level granularity (bSPARQ). We can rethink how to distinguish bit-level granularity from just "sparsity", thereby avoid misleading some readers.
>
>
> * Q: "For bSPARQ, what about trimming the bit-width with non-consecutive but equally spanned digits, e.g., 4-bit at 1,3,5,7 or 2,4,6,8 digits from the 8-bit representation?”
> A: This option is another “2opt” variation which we can empirically measure if it works better than our MSB-LSB 2opt configuration. In a broader sense, we can see if there are particular patterns which are more likely to occur than others. For example, in the 5opt configuration, instead of 5 consecutive 4-bit windows, 5 different 4-bit patterns which are not necessarily consecutive. We will definitely put it in our to-do list for future work. Thank you for this idea.

---

> > ### Comment · Reviewer_8ZWQ · 2021-08-25
> > **Additional comments**
> >
> > Thanks for the response. Here are some additional comments:
> >
> > - Q: "The paper has missing some recent efforts on post-training quantization listed below [1,2,3,4]. These works largely follow uniform quantization and are therefore friendly to general hardwares."
> >
> >   A: Thank you for these pointers. Specifically, LAPQ [4] does not employ per-channel quantization, which makes the comparison not fair. As for the rest of the papers, we will check them and add them to our comparison table.
> >
> >   Comment: I still think it would be better to tone down the advantage "SPARQ achieves minor accuracy degradation, 2× speedup 17 over widely used hardware architectures, and a practical hardware implementation. ". As pointed out by the authors, this is the potential speed-up that has not been verified, thus it could be mis-leading if this is claimed as practical advantage.
> >
> > - Q: "It would be more comprehensive to compare with the missing recent works [1,2,3,4]? Especially, BRECQ[2] also considers the low bit post-training quantization on ResNet-18 and ResNet-50. It would be also convenient to compare with these approaches with absolute accuracy values reported."
> >
> >   A: Thank you for that pointer. We will gladly add BRECQ to our comparison tables. For example, BRECQ achieves relative accuracy degradation of 0.53%, 1.79%, and 6.72% for ResNet-18 with 4/32, 3/32, and 2/32, respectively, whereas SPARQ achieves 0.07%, 0.21%, and 1.64%.
> >
> >   Comment: The comparison with BRECQ is fine. "BRECQ achieves relative accuracy degradation of 0.53%, 1.79%, and 6.72% for ResNet-18" -> It seems the degradation number are not true according to their results (https://arxiv.org/pdf/2102.05426.pdf)
> >
> > - Unanswered: "What about the time consumption on hardware to search the most significant bits in bSPARQ (L111)? ". Will this affect the potential 2x speed-up for the proposed approach?

---

> > > ### Author Response · Authors · 2021-08-25
> > > **Response to Reviewer 8ZWQ**
> > >
> > > We would like to thank the reviewer again for the constructive criticism.
> > >
> > > * Q: I still think it would be better to tone down the advantage "SPARQ achieves minor accuracy degradation, 2× speedup 17 over widely used hardware architectures, and a practical hardware implementation. ". As pointed out by the authors, this is the potential speed-up that has not been verified, thus it could be misleading if this is claimed as practical advantage.
> > > A: We appreciate the reviewer's feedback and agree with him. We will tone down this advantage throughout our manuscript in the next revision. It does not change SPARQ contribution to prior work.
> > >
> > > * Q: The comparison with BRECQ is fine. "BRECQ achieves relative accuracy degradation of 0.53%, 1.79%, and 6.72% for ResNet-18" -> It seems the degradation number are not true according to their results (https://arxiv.org/pdf/2102.05426.pdf).
> > > A: We derived the relative accuracy degradations given BRECQ results in Table 2.
> > > For example, BRECQ 4/32 achieves 70.7, which is relatively 0.53% lower than their 71.08 baseline (calculation: 100 * (71.08 - 70.70) / 71.08).
> > >
> > > * Q: "What about the time consumption on hardware to search the most significant bits in bSPARQ (L111)? ". Will this affect the potential 2x speed-up for the proposed approach?
> > > A: We would like to apologize for missing this question.
> > > We estimate that the additional time searching for the most significant bits with bSPARQ is negligible. Leading zero counters (LZCs) are known basic blocks in logic design; for example, a 4-bit LZC, as in the 5opt configuration, requires a small number of additional logic gates [1]. Also, as we mention in the manuscript, bSPARQ (and vSPARQ) can be performed outside the computation engines; therefore, they are less prone to impact the critical path.
> > >
> > > [1] Milenkovic, Nebojsa Z. et al. "Modular design of fast leading zeros counting circuit." Journal of Electrical Engineering 66.6 (2015): 329.

---

### Official Review · Reviewer_f1rB · 2021-07-18

**Rating:** 5
**Confidence:** 4

**Summary:**

This paper proposes a post-training sparsity-aware quantization algorithm, SPARQ, for neural networks. SPARQ leverages the bitwise sparsity by skipping leading zero-value during quantization. SPARQ also leverages the elementwise sparsity by dynamically increase the quantization precision if one in the activation pair is zero. This paper also presents a practical hardware implementation for supporting the SPARQ. The experiments show a 0.18% accuracy drop in 4-bit quantized ResNet50 on ImageNet.

**Limitations And Societal Impact:**

The authors have adequately addressed the limitations and societal impact.

**Main Review:**

Strength:
- The writing is clear. The paper is well structured and easy to follow.
- The paper presents the corresponding hardware design and analyzes the extra area cost for supporting the proposed quantization method.

Weakness:
- The evaluation is mainly conducted on large-scale neural networks such as ResNet and GoogleNet. It is unclear how SPARQ performs on lightweight networks such as SqueezeNet and MobileNets.
- The evaluation lacks an ablation study. It is unclear how much improvement comes from bSPARQ and how much comes from vSPARQ.
- Each bSPARQ quantized value requires additional index bits. For example, 4-bit bSPARQ with 5opt is actually 7-bit quantization, and 2-bit with 7opt is 5-bit quantization. It remains questionable whether it is fair to compare against other 4-bit quantization methods. Moreover, it is unclear how speedup is calculated. According to the roofline model, most neural network computation is memory-bounded. Since the memory footprint does not decrease much, what is the real benefit of SPARQ in terms of both latency and energy? Even though it has been stated in the limitation section in the paper, it is a severe problem of the proposed method.

**Time Spent Reviewing:**

1

---

> ### Author Response · Authors · 2021-08-07
> **Response to Reviewer f1rB**
>
> We would like to thank the reviewer for the helpful and valuable feedback and suggestions. We would like to address the reviewer comments about the paper’s weaknesses:
>
>
> * Q: “The evaluation is mainly conducted on large-scale neural networks such as ResNet and GoogleNet. It is unclear how SPARQ performs on lightweight networks such as SqueezeNet and MobileNets.”
> A: Following the reviewer suggestion, we enabled SqueezeNet (top-1 of 58.09, from PyTorch) with SPARQ. We measured the following relative accuracy degradation results with rounding for 5opt, 3opt, and 2opt, respectively: 0.8%, 1.05%, 8.24%. LBQ, for example, achieves relative degradation of 2.96%. We also have lower bit widths results. We will gladly add SqueezeNet results to the relevant tables.
>
>
> * Q: “The evaluation lacks an ablation study. It is unclear how much improvement comes from bSPARQ and how much comes from vSPARQ.”
> A: We thank the reviewer for this suggestion. We enabled an additional argument in our code which disables vSPARQ, and measured all operating points with only bSPARQ. We will gladly add the numerical results and conclusions into a revision and update our code in the repository. These are our conclusions:
>     * With 5opt, vSPARQ impact is minor, since bSPARQ incurs relatively small quantization noise. If we remove vSPARQ from our hardware implementation (the two top multiplexers and MuxCtrl in Figure 2), we measure 10% area overhead decrease for the 5opt implementations in Table 4.
>     * Given 5opt results, the 3opt configuration becomes obsolete, since for almost the same hardware, 5opt achieves better accuracies.
>     * vSPARQ comes into play for 2opt and lower bit widths (Table 5), in which the relatively high quantization noise of a quantized element can be mitigated by leveraging sparsity. For example, ResNet-101 top-1 accuracy with 2-bit vSPARQ is 75.33%, whereas without vSPARQ it is 74.49%.
>
>
> * Q: “Each bSPARQ quantized value requires additional index bits. For example, 4-bit bSPARQ with 5opt is actually 7-bit quantization, and 2-bit with 7opt is 5-bit quantization. It remains questionable whether it is fair to compare against other 4-bit quantization methods. Moreover, it is unclear how speedup is calculated. According to the roofline model, most neural network computation is memory bounded. Since the memory footprint does not decrease much, what is the real benefit of SPARQ in terms of both latency and energy? Even though it has been stated in the limitation section in the paper, it is a severe problem of the proposed method.”
> A: As for the reviewer’s first concern, regarding whether it is fair to compare our method against other 4-bit quantization methods – from a computational point of view, we argue it is a fair comparison, since all these methods are based on 4b-8b multipliers (line 234).
> As for the reviewer’s second concern, regarding speedup, latency, and energy – the potential speedup of, for example, 4b-8b models over 8b-8b models is 2x, given the same area, give or take, since an 8b-8b multiplication is equivalent to two 4b-8b multiplications (Equation 3). Latency, therefore, decreases as well. Energy, from a system point of view, may also decrease, by mitigating static power consumption; or given the same performance, frequency can be reduced, and thereby voltage can be reduced. All these things, however, are the result of most quantization works.
>
> SPARQ's novelty is that it is a quantization method that leverages sparsity: (1) in the representation granularity (vSPARQ) and (2) in bit-level granularity (bSPARQ). Indeed, the current SPARQ implementation requires some control overheads, which translate to memory footprint; however, we hope that this concept will trigger future works that mitigate the additional memory overheads (we have an on-going project for that as well).

---

> > ### Comment · Reviewer_f1rB · 2021-08-28
> > **Thanks for the response**
> >
> > I appreciate these clarifications from the authors. However, my concerns are not yet addressed.
> > - The experiments on SqueezeNet are helpful. However, it would be better to see how SPARQ performs on MobileNet-like networks since depthwise separable convolution is more difficult to be quantized in general.
> > - Regarding the comparison against 4-bit quantization methods, the response is not convincing. Consider an already quantized 8-bit activation. A 4-bit bSPARQ with 5opt requires 4 bits for the quantized value and 3 bits for trimming the leading zeros (or 3 bits for left-shift). Thus bSPARQ can be considered as 7-bit "float" with a 3-bit exponent part and a 4-bit mantissa. Even from the computational point of view, the MAC on bSPARQ needs additional left shifting which is not required by conventional integer MAC.
> > - Regarding the claim of 2x speedup, the authors completely ignore the effect of data movement on the latency. 2x fewer BitOps do not yield a 2x speedup. For example, though INT4 doubles the computation throughput compared to INT8 on NVIDIA tensor core[1],  the end-to-end INT4 quantized ResNet50 is only 1.2X speedup [2]. According to [3], after quantization, the neural network may become memory-bounded. The memory issue could be the problem when inferencing the SPARQ-quantized models.
> >
> > [1] Figure 8. https://images.nvidia.com/aem-dam/Solutions/design-visualization/technologies/turing-architecture/NVIDIA-Turing-Architecture-Whitepaper.pdf
> >
> > [2] https://discuss.tvm.apache.org/t/rfc-tensorcore-int4-end-to-end-inference/7534
> >
> > [3] https://www.xilinx.com/support/documentation/white_papers/wp521-4bit-optimization.pdf

---

### Official Review · Reviewer_J1hE · 2021-07-19

**Rating:** 7
**Confidence:** 5

**Summary:**

The authors propose a post-training quantization scheme comprising two orthogonal ideas: bSPARQ and vSPARQ.

bSPARQ a dynamic quantization technique. Instead of rounding values to the required bitwidth (equivalent to keeping the MSBs), the technique keeps the most significant consecutive non-zero bits (i.e. the sequence of bits starting from the first non-zero bit). This is equivalent to making a power-of-two adjustment to the scale factor for each individual value.

vSPARQ is a dynamic sparsity technique meant for activations. First, adjacent pairs of activatons are grouped (e.g. [x1, x2, x3, x4] -> [x1x2, x3x4]). In each pair, if one of the two activations is zero, the other can make use of its compute resources. The idea is detailed in Equation (2) and can be realized in hardware with the architecture in Figure 2.

The authors demonstrate SOTA accuracy results on CNN image classification for very low activation bitwidths (4/3/2-bit activations and 8-bit weights). However, the technique also incurs significant area overhead: 22% for the 3opt design point.

**Limitations And Societal Impact:**

Yes

**Main Review:**

This is a very interesting paper that at first glance combines two unrelated techniques: bSPARQ shifts each value and vSPARQ exploits dynamic sparsity. However, both techniques can make use of a shifter after the multiplier (see Figure 2) which is what makes combining them a compelling idea. Overall a strong paper with solid results on both accuracy and hardware.

Some comments/critiques of the paper:
1. The overhead of the control bits is significant (3 bits per 4-bit activations with the 3-opt configuration). The area results on the multiplier unit shows that this is not an issue for compute, but for routing and memory it feels like a big problem. In particular, the paper doesn't describe the datapath at the output of the conv/matmul unit, which must re-quantize the wide accumulator outputs back to 4-bit for the next layer. This datapath will likely be significantly larger due to the overhead bits and the logic to generate them. I think this is what may kill the idea in practice.

2. The accuracy comparison in Table 2 is a bit misleading. ACIQ, for example, incurs no area overhead. This should be pointed out in Table 3 (which specifically only mentions SySMT to make the comparison look better).

3. The authors should cite https://arxiv.org/abs/1910.06909 which proposes a very similar idea to vSPARQ, with more analysis.

4. The hardware evaluation methodology section in the supplements should be moved to main paper to assure readers that a solid hardware evaluation was done.

**Time Spent Reviewing:**

2

---

> ### Author Response · Authors · 2021-08-07
> **Response to Reviewer J1hE**
>
> We would like to thank the reviewer for the helpful and valuable feedback and suggestions. We would like to address the reviewer’s comments and critiques:
>
>
> * Q: “The overhead of the control bits is significant (3 bits per 4-bit activations with the 3-opt configuration). The area results on the multiplier unit shows that this is not an issue for compute, but for routing and memory it feels like a big problem. In particular, the paper doesn't describe the datapath at the output of the conv/matmul unit, which must re-quantize the wide accumulator outputs back to 4-bit for the next layer. This datapath will likely be significantly larger due to the overhead bits and the logic to generate them. I think this is what may kill the idea in practice.”
> A: Indeed, there are additional overheads due to control bits. However, we argue the datapath is not that different from any 8-bit accelerator. The re-quantization process is not dramatically different from other accelerators, in which the quantized output activations are re-scaled from 32-bit back to a lower bit width. Also, we argue that the area overhead due to the logic that generates the control bits is negligible in the systolic array case, since it can be done outside systolic array in a lower frequency (i.e., it is computed for pairs of activations after hundreds or thousands of MAC operations). In the context of Sparse Tensor Cores, we implemented that logic, since we couldn't claim this logic is negligible anymore (Section 5.4). For the 5opt configuration, we estimate the area overhead as 17% from the dot-product unit size (Figure 4). Given the additional multiplexers in Sparse Tensor Core (Figure 5), the area overheads would be smaller.
>
>
> * Q: “The accuracy comparison in Table 2 is a bit misleading. ACIQ, for example, incurs no area overhead. This should be pointed out in Table 3 (which specifically only mentions SySMT to make the comparison look better).”
> A: We had no intention misleading the readers; on the contrary, the column 2x4b-8b (Table 3) was added to represent the “conventional” quantization schemes, such as ACIQ. In Section 6, we also admit that “compared with native 4b-8b quantizations, our hardware implementation incurs some overhead, due to the additional hardware required for the multiplier and shift logic” (line 294). Having said all that, we will point it out more clearly in a revision.
>
>
> * Q: “The authors should cite https://arxiv.org/abs/1910.06909 which proposes a very similar idea to vSPARQ, with more analysis.”
> A: Thank you for that pointer, we will gladly do so.
>
>
> * Q: “The hardware evaluation methodology section in the supplements should be moved to main paper to assure readers that a solid hardware evaluation was done.”
> A: Thank you for that as well. We will try to squeeze the evaluation methodology to the main paper instead of being at the supplementary material.

---

### Decision · Program_Chairs · 2021-09-27

**Decision:**

Accept (Poster)

**Comment:**

This paper explores the activation sparsity at both bit-level and numerical-level for 8-bit post-training quantization. For bit-level, a dynamic quantization technique is proposed to dynamically examine and trim leading zero-bits, which enabling a multi-scale quantization and a 2x4bit-8bit MAC unit. For numerical level, the paper proposed to skip a pair of activations when one of them is zero. The techniques are evaluated on popular CNN vision models and dataset, achieving baseline accuracy and potentially 2x speedup with some reasonable overhead.

Overall, the paper is strong. The techniques introduced are interesting and novel. The experiments are sound, and results are solid. However, it would have been helpful if the author could describe more clearly on the hardware benefits, which seems to be confusing to some reviewers. In addition, the hardware evaluation methodology in appendix should be moved to the main paper as suggested by reviewers, plus more detailed explanation on the overhead of control logic.

In short, this paper is recommended for acceptance.